# Myokine Secretion following an Aerobic Exercise Intervention in Individuals with Type 2 Diabetes with or without Exercise Resistance

**DOI:** 10.3390/ijms25094889

**Published:** 2024-04-30

**Authors:** Léa Garneau, Erin E. Mulvihill, Steven R. Smith, Lauren M. Sparks, Céline Aguer

**Affiliations:** 1Faculty of Medicine, Department of Biochemistry, Microbiology and Immunology, University of Ottawa, Ottawa, ON K1H 8M5, Canada; lgarn072@uottawa.ca (L.G.); emulvihi@uottawa.ca (E.E.M.); 2Institut du Savoir Montfort, Ottawa, ON K1K 0T2, Canada; 3University of Ottawa Heart Institute, Ottawa, ON K1Y 4W7, Canada; 4Translational Research Institute for Metabolism and Diabetes, AdventHealth Orlando, Orlando, FL 32804, USA; steven.r.smith@adventhealth.com (S.R.S.); lauren.sparks@adventhealth.com (L.M.S.); 5Faculty of Medicine and Health Sciences, Department of Physiology, McGill University–Campus Outaouais, Gatineau, QC J8V 3T4, Canada; 6Faculty of Health Sciences, School of Human Kinetics, University of Ottawa, Ottawa, ON K1N 6N5, Canada

**Keywords:** myokines, type 2 diabetes, obesity, aerobic exercise, training intervention, skeletal muscle, exercise resistance, inflammation

## Abstract

Type 2 diabetes (T2D) is characterized by muscle metabolic dysfunction that exercise can minimize, but some patients do not respond to an exercise intervention. Myokine secretion is intrinsically altered in patients with T2D, but the role of myokines in exercise resistance in this patient population has never been studied. We sought to determine if changes in myokine secretion were linked to the response to an exercise intervention in patients with T2D. The participants followed a 10-week aerobic exercise training intervention, and patients with T2D were grouped based on muscle mitochondrial function improvement (responders versus non-responders). We measured myokines in serum and cell-culture medium of myotubes derived from participants pre- and post-intervention and in response to an in vitro model of muscle contraction. We also quantified the expression of genes related to inflammation in the myotubes pre- and post-intervention. No significant differences were detected depending on T2D status or response to exercise in the biological markers measured, with the exception of modest differences in expression patterns for certain myokines (IL-1β, IL-8, IL-10, and IL-15). Further investigation into the molecular mechanisms involving myokines may explain exercise resistance with T2D; however, the role in metabolic adaptations to exercise in T2D requires further investigation.

## 1. Introduction

Type 2 diabetes (T2D) is a chronic, non-communicable disease associated with poor nutritional habits, reduced physical activity, and increased sedentary behavior [1,2,3]. In patients with T2D, some of the earliest metabolic manifestations of the disease are the development of skeletal muscle insulin resistance and mitochondrial dysfunction [4,5]. Exercise interventions are a popular addition and alternative to traditional pharmacological treatments for the management of T2D, as they have been shown to increase skeletal muscle mitochondrial oxidative capacity and improve insulin sensitivity [6,7]. The AMP-activated protein kinase (AMPK) pathway is central to muscle signaling during exercise, since it serves as an energy sensor responding to the rise in the AMP/ATP ratio that occurs during muscle contraction [8]. Muscle contraction causes AMPK phosphorylation and the consequent downstream activation of peroxisome proliferator-activated receptor gamma coactivator 1 alpha (PGC1α), an important modulator of mitochondrial dynamics that enhances mitochondrial oxidative capacity (OXPHOS) [9]. PGC1α can also be activated by sirtuins in response to exercise, causing remodeling of the electron-transport chain complexes to enhance OXPHOS efficiency [10,11]. In addition to these intracellular signaling events, the metabolism of skeletal muscle cells is regulated by extracellular signaling mechanisms during exercise.

Indeed, the skeletal muscle is an important secretory organ, releasing peptides called myokines in the muscle interstitium during muscle contraction [12]. Myokines can be regulated both at the transcript and protein levels by acute and chronic exercise, and they have been shown to greatly impact the regulation of insulin sensitivity and mitochondrial function in skeletal muscle [13]. For example, interleukin (IL)-6, IL-10, IL-13, IL-15, IL-18, brain derived neurotrophic factor (BDNF), fibroblast growth factor 21 (FGF21), and secreted protein acidic and rich in cysteine (SPARC) act locally in muscle cells by improving mitochondrial function, insulin-independent glucose uptake, and/or insulin sensitivity [14,15,16,17,18,19,20,21,22,23,24,25,26,27]. Many of these myokines modify skeletal muscle metabolism through the AMPK pathway following exercise [20,24,27,28]. As such, exposing L6 rat skeletal muscle cells to IL-15 at physiological concentrations enhances mitochondrial function in parallel to the activation of the AMPK pathway and increased glucose uptake [29]. Some myokines are thought to play an endocrine role in regulating metabolic adaptations to exercise. IL-6 has been shown to influence insulin secretion [30,31,32,33], and IL-13 can positively modulate insulin secretion by improving β-cells function and/or limiting cytokine-induced β-cell apoptosis [34]. Conversely, FGF21 treatment improved hepatic and whole-body insulin sensitivity in a mouse model of diet-induced obesity [35]. FGF21 has also been shown to regulate lipolysis in adipose tissue [36]. IL-15 treatment in adipocytes results in an increased secretion of adiponectin [37], an adipokine known to promote insulin sensitivity, as well as increased mobilization and catabolism of circulating lipids [38]. Therefore, myokines are important modulators of both skeletal muscle and whole-body metabolism in response to an exercise intervention.

Some subjects with T2D experience no improvements in glucose homeostasis in response to exercise interventions, a phenomenon termed ‘exercise resistance’. For example, after an exercise intervention protocol of 12–16 weeks of aerobic moderate-intensity continuous training in a cohort of around one hundred subjects with pre-diabetes or T2D, roughly one-third of participants showed no improvements in glucose control [39]. Similar results were found in women with impaired glucose tolerance after a 12-week high-intensity interval training or resistance training intervention [40]. This supports the notion of non-responders to glucose-control improvements with different types of interventions. Moreover, in vivo and ex vivo assessment of mitochondrial function via the measurement of the phospho-creatine (PCr) recovery rate in patients with T2D following a 10-week exercise intervention established that the lack of improvement in insulin sensitivity correlates with the absence of changes in mitochondrial function [41]. The primary outcome of the RESIST study (NCT01911104) showed that participants who qualified as non-responders had higher expression patterns of genes linked to antioxidants, mitochondrial metabolism, and insulin signaling at baseline compared to responders [42]. These epigenetic molecular regulatory mechanisms were maintained in biopsy-derived primary muscle cells cultured in vitro, suggesting a cell-autonomous process.

Many other factors, such as duration of T2D, age, genetics, and epigenetics, contribute to the exercise response in patients with T2D, regardless of exogenous factors, such as the type and duration of the exercise intervention [43]. Although myokine secretion is intrinsically altered in patients with T2D [44], the myokine profiles of responders and non-responders to the beneficial effects of exercise have not yet been compared after a training intervention. Altered myokine secretion in response to acute and chronic exercise could, therefore, contribute to the impaired improvements in glucose homeostasis and mitochondrial function in patients with T2D who are non-responsive to an exercise intervention.

As myokines are mainly cytokines that can be secreted by other tissues, the measurement of their levels in plasma or serum does not adequately represent the secretion of myokines by skeletal muscle. That said, when isolated from the muscle biopsies of patients with T2D and cultured in vitro, human primary skeletal muscle cells maintain their insulin resistance, mitochondrial dysfunction, and altered myokine secretion [44,45,46,47,48]. Isolated muscle cells can be differentiated into myotubes that resemble mature muscle cells. This model, using samples obtained directly from clinical participants, is ideal for the study of myokines secreted specifically by skeletal muscle cells during T2D without the interference of surrounding tissues [49]. In addition, myotubes can be subjected to electrical pulse stimulations (EPS), which mimic muscle contraction in vitro [50]. EPS has been shown to induce metabolic adaptations similar to those observed in muscle after exercise, while also increasing myokine secretion [51,52]. Most studies using EPS with human primary muscle cells employed long durations of stimulations (4–48 h) at low frequencies (≤2 Hz), resulting in improved oxidative functions, mitochondrial biogenesis, glucose uptake, insulin sensitivity, and increased release of myokines in the cell-culture supernatant [51,52,53,54,55,56]. That being said, these stimulation conditions do not translate into muscle contraction in vivo, as the stimulus is considerably longer than an average bout of exercise would last. To mimic acute muscle contraction, Li et al. treated C2C12 mouse muscle cells with 1 h of EPS at 20 V, 1 Hz frequency with 24 ms impulses every 976 ms (24 ms on the second), resulting in increased cell-surface glucose transporter GLUT4 localization through AMPK and downstream TBC1D1 and Akt Substrate 160 (AS160 also named TBC1 Domain Family Member 4, TBC1D4) signaling [57]. C2C12 myotubes have a greater EPS contractile activity than human primary muscle cells because of their more rounded morphology [58]. This characteristic of human myotubes warrants validation of these stimulation conditions prior to any inference between in vivo exercise and in vitro myotube contraction in clinically derived cell lines.

This study is a secondary analysis of the randomized controlled trial NCT01911104, which focused on the epigenetic mechanisms occurring in skeletal muscle and contributing to exercise resistance in patients with T2D. We sought to determine if myokine signaling adaptations to exercise and muscle contraction differed according to T2D status, and/or as a function of the ability to respond to an exercise intervention in the case of participants living with T2D as assessed by their rate of muscle PCr recovery. To this end, we measured myokines in the serum samples of participants pre- and post-intervention, in the cell-culture supernatant of biopsy-derived human primary muscle cells collected pre- or post-intervention and stimulated or not with an in vitro model of muscle contraction (EPS). We also measured the expression of myokines in the isolated muscle cells pre- and post-intervention. We hypothesized that local myokine secretion (measurements in cell-culture supernatant samples) might be dysregulated in the group of patients with T2D who do not manifest metabolic improvements with exercise training in comparison to those who benefit from the intervention.

## 2. Results

### 2.1. Patient Characteristics

The anthropometric and metabolic characteristics of the participants in this sub-study can be found in Table 1; full cohort characteristics have been previously published [41]. No differences were found between groups for age, BMI, fat mass, lean mass, VO_2max,_ and adherence to the program (No T2D 79.0%; T2D responders 76.3%; T2D non-responders 76.8%). The apparent higher mean age in the T2D responders group and mean BMI in the No T2D group are due to a single high value in each group, and the *p*-values were not significant (*p* = 0.1016 and *p* = 0.5109, respectively). There were no significant differences amongst the groups pre- compared to post-intervention for BMI, fat mass, and VO_2max_. Glycated hemoglobin (HbA1c) levels were higher in the two groups of participants with T2D compared to the no T2D group independently of the intervention status (*p* = 0.0007). The effect of the intervention was significant on HbA1c values (*p* = 0.0342), and a significant interaction was detected between the groups and the effect of the intervention (*p* = 0.0131). In the group of patients with T2D classified as non-responders to the exercise intervention, HbA1c levels were significantly higher post-intervention compared to pre-intervention (*p* = 0.0012). This effect on HbA1c levels is associated with the interruption of glucose-lowering treatment in participants with T2D. An effect of the intervention was found for changes in lean mass across all groups (*p* = 0.0013), and the increase reached significance in the T2D non-responders group in the post hoc test (*p* = 0.0075). Although M-values during the hyperinsulinemic-euglycemic clamp did not vary significantly pre- compared to post-intervention, they were significantly different between groups (*p* = 0.0060). Pre-intervention, the M-value was lower only in the T2D responders group compared to the no T2D (*p* = 0.0309), while post-intervention, the M-value was lower in both the T2D responders (*p* = 0.0120) and the T2D non-responders (*p* = 0.0016) groups compared to the participants with no T2D. A significant interaction was found between the effects of the group and the intervention, with the intervention showing no significant effect on PCr recovery rate in the group of participants with no T2D and the T2D responders group, while a decrease occurred in the T2D non-responders group (*p* = 0.0089). These results are in alignment with the findings from the primary outcome study, which showed a positive correlation between insulin sensitivity (M-value) and PCr recovery rate at the pre-intervention stage in the two groups of participants with T2D [41]. Both measures of energy metabolism were increased in the group of participants with T2D who qualified as non-responders in comparison to responders before the start of the intervention. These results were accompanied by a favorable epigenetic and transcriptional profile of muscle metabolic function in the non-responders group compared to patients with T2D who qualified as responders.

### 2.2. Serum Cytokine Concentrations following the Aerobic Exercise Intervention

The training intervention did not affect the serum concentration of the cytokines measured in this participant population (Figure 1A–G). There were no significant differences between groups for SPARC, FGF21, IL-10, IL-15, and IL-18 serum concentrations, irrespective of the intervention status, but there was a trend towards lower circulating IL-6 levels in the T2D non-responders, although the post hoc test was not significant (*p* = 0.0982). Serum IL-8 concentrations were different across groups, irrespective of intervention status (*p* = 0.0074), with a trend towards an increase in the T2D responders group in comparison to the no T2D group (*p* = 0.0673), and a significantly higher concentration of serum IL-8 in the T2D non-responders group compared to the no T2D group (*p* = 0.0067). IL-13 was mostly undetected in the serum samples across all groups and independently of intervention, preventing the possibility of further data analysis.

### 2.3. Myokine Secretion by Primary Human Muscle Cells in Response to the Intervention

As the cytokines measured in the serum samples (Figure 1) could originate from numerous different tissues, we also measured those secreted by the primary muscle cells derived from skeletal muscle biopsies of the vastus lateralis collected pre- and post-intervention in a subset of the study participants to infer muscle-specific secretion of the myokines. We measured myokines in the cell-culture supernatant of the biopsy-derived muscle cells exposed to the myotubes for 1 h and found no differences in myokine concentrations between pre- and post-intervention (Figure 2A–F). Most of the candidate myokines measured were detected in the cell-culture supernatant (SPARC, IL-6, IL-8, IL-10, IL-15, and IL-18), but FGF21, BDNF, and IL-13 were mostly undetected independently of the group or intervention status of the participants from which the cells originated.

### 2.4. Correlation between Myotube Secretion and Serum Myokine Concentrations

Serum samples collected from participants for which human primary muscle cells were available were used to assess any potential correlation between myokine secretion in the circulation (peripheral) and primary muscle cells in culture (local). Data from participants across all groups pre- and post-intervention were pooled for the analyses. No significant correlations were found between serum and cell-culture supernatant concentrations for any of the cytokines/myokines assessed (SPARC, IL-6, IL-8, IL-10, IL-15, and IL-18; Figure 3A–F).

### 2.5. Effects of EPS Treatment on Primary Muscle Cell Molecular Signaling Events

To mimic acute muscle contraction, the primary myotubes derived from muscle biopsies obtained from the participants pre- and post-intervention were subjected to an EPS treatment (S). Non-stimulated (NS) and electrode exposure without current (E) were used as control conditions to determine the effects of muscle cell contraction per se as opposed to the potentially pro-inflammatory effect of the carbon electrode on the myotubes [59].

#### 2.5.1. AMPK-Pathway Activation

As mentioned previously, the AMPK signaling pathway is central to metabolic adaptations during acute muscle contraction. Upon muscle contraction and phosphorylation of AMPK, downstream phosphorylation of Acetyl-CoA Carboxylase (ACC) activates fatty acid oxidation, while phosphorylation of AS160 contributes to GLUT4 translocation to the plasma membrane. In the cell lines from the no T2D group collected post-intervention, the EPS condition (S) showed significantly higher p-AMPK compared to the NS condition (Figure 4A,B, *p* = 0.0179). In the cell lines from the T2D non-responders group collected pre-intervention, the electrode treatment (E) resulted in more p-AMPK than the NS condition (Figure 4A,B, *p* = 0.0332). In the myotubes from the participants in the no T2D group pre-intervention, and in the myotubes from the T2D responders group post-intervention, the EPS condition (S) caused significantly more p-AS160 compared to the NS condition (Figure 4A and Figure 4C, *p* = 0.0035 and *p* = 0.0220, respectively). Both the EPS treatments and the study group from which the myotubes were derived (effect of group) had a significant effect on the regulation of pACC levels (Figure 4A and Figure 4D, *p* = 0.0057 and *p* = 0.0205, respectively). The EPS condition (S) significantly increased p-ACC in the myotubes from the participants of the no T2D group at both pre- and post-intervention (Figure 4A and Figure 4D, *p* = 0.0047 and *p* = 0.0032, respectively).

#### 2.5.2. Myokine Secretion

To determine the potential effects of EPS treatments on myokine secretion, the biopsy-derived primary cell cultures collected pre- and post-intervention were submitted to the three different conditions (NS, E, and S), and the myokine concentrations in the cell-culture supernatant of each cell line were assessed. Myokine secretion in vitro was compared between the EPS treatments for each cell line relative to the NS condition.

No effect of electrode exposure (E) was observed on the myokine secretion by the primary human muscle cells obtained from the participants in comparison to the NS condition (Figure 5A–F). There was a significant effect of groups on SPARC secretion by the myotubes (*p* = 0.0322), but no significant differences were detected in the post hoc tests (Figure 5A).

Treatment of the myotubes derived from the participants in all groups with the 1 h EPS protocol (stimulated; S) did not significantly alter myokine secretion in comparison to the non-stimulated (NS) condition (Figure 6A–F). Most targeted myokines were detected in the cell-culture supernatant (SPARC, IL-6, IL-8, IL-10, IL-15, and IL-18), but some were mostly undetected, and their secretion pattern could not be analyzed (BDNF, FGF21, and IL-13).

### 2.6. Regulation of Gene Expression in Human Primary Myotubes following the Intervention

Gene-expression analyses of the NS condition of myotubes from the participant groups revealed a regulatory effect of both the exercise intervention and the groups on the expression of genes related to inflammation (Figure 7A–H). The level of mRNA expression increased between pre- and post-intervention in the T2D responders group for IL-1β (Figure 7B, *p* = 0.0116) and IL-8 (Figure 7D, *p* = 0.0114), as well as in the no T2D and T2D non-responders groups for IL-15 (Figure 7G, *p* = 0.0023 and *p* = 0.0007, respectively). An interaction between the effect of the intervention and the group was detected for IL-10 (*p* = 0.0270) and IL-15 (*p* = 0.0084) (Figure 7E,G). Levels of IL-10 mRNA were significantly decreased post- compared to pre-intervention in the T2D non-responders group (*p* = 0.0459), while IL-10 expression trended towards an increase post-intervention in the two other groups with no significance observed (Figure 7E). Although IL-15 mRNA levels significantly increased in the no T2D (*p* = 0.0023) and T2D non-responders groups (*p* = 0.0007), there was no change in the T2D responders group (Figure 7G).

## 3. Discussion

Aerobic exercise training is a key non-pharmacological method for the management of impaired glucose homeostasis in patients with T2D. Unfortunately, some patients living with diabetes do not benefit from the metabolic improvements associated with increased aerobic exercise. We sought to determine whether myokine signaling was a contributing factor to the mechanisms of exercise resistance in patients with T2D. We found no effect of either T2D or the capacity to respond to an exercise intervention on peripheral or local signaling adaptations at the protein level of target cytokines/myokines using serum samples and biopsy-derived primary muscle cells collected from the study participants pre- and post-exercise intervention. However, the expression-pattern variations in response to the training intervention of some of the candidate myokines quantified at the mRNA level differed according to T2D and responder or non-responder status. To our knowledge, this is the first study to examine and compare local and peripheral regulation of myokine signaling in participants with obesity with or without T2D in response to an aerobic exercise intervention protocol while accounting for the ability of the patients with T2D to benefit from the intervention at the mitochondrial metabolic level. An important consideration to take into account regarding our study before discussing the results is the absence of a group of participants not undergoing the exercise training intervention. Therefore, all analyses pertaining to pre- and post-intervention states cannot be controlled for the variations in all factors assessed as a function of the natural evolution of the parameters over the 10-week time course of the observations. Moreover, the original study design did not include a group of sedentary participants with a BMI in the “healthy” range, but with characteristics similar to the groups of participants with obesity. This limitation eliminates the potential of detecting any effects of obesity on muscle signaling adaptions. Finally, another important caveat of this study is the small number of participants per group, which causes a reduction in the power of the analyses performed on the data. This limitation is a consequence of the nature of the study, which serves as a secondary analysis of samples from a previously published broader clinical trial, for which we did not have access to the entire study sample pool.

We assessed the levels of circulating cytokines that are also known to be secreted by skeletal muscles in serum samples collected from the participants at rest. The only significant difference we found in the participant population was an increase in serum IL-8 levels in the two groups of patients with T2D in comparison to those without, irrespective of their response to exercise-induced metabolic adaptations. This aligns with the previous characterization of circulating IL-8 levels in patients with T2D found in the literature [13,60]. That being said, others have shown that some of the cytokines (i.e., SPARC, IL-6, IL-10, IL-13, IL-15, and IL-18) we quantified in our cohort can be altered in the circulation of patients with T2D, as we reviewed in 2019 [13]. Since we measured circulating cytokines in serum and most of these other studies quantified cytokines in plasma, the medium assessed could explain the inconsistencies in our findings compared to others. A recent meta-analysis examining the correlation between variations in circulating levels of certain so-called exerkines found that FGF21, IL-6, and IL-10, among others, could serve as potential biomarkers to assess the effectiveness of an exercise protocol on improvements in glucose homeostasis in patients with T2D [61]. For most cytokines, variations in response to either acute or chronic exercise can be better detected in plasma samples [62], which could explain why we saw no effect of the training intervention on the circulating levels of the cytokines.

The analysis of cell-culture supernatant from primary muscle cells derived from the participants pre- and post-intervention revealed no differences associated with either intervention status or group. This finding contradicts those of others, who found that primary human muscle cells from patients with obesity and T2D secreted myokines differently than participants without T2D [44]. Indeed, they found a higher secretion of IL-6, IL-8, and IL-15 in the muscle cells of patients with T2D in comparison to the no-T2D group. This discrepancy could be explained by the fact that our study groups included participants with similar anthropometric values, whereas the group of patients with T2D in the Ciaraldi et al. study had a significantly higher BMI than their insulin-sensitive counterparts. Another important factor is that the conditioned medium containing the myokines was collected after only one hour of exposure to the cells, while the other team measured the myokines after 24 h and 48 h of exposure. Perhaps a longer time elapsing before collection would have allowed for better quantification of the secreted myokines. Moreover, we were unable to normalize the protein concentration of myokines in the cell-culture medium to the total protein content of their respective lysates, as the samples were used for RNA isolation. The resulting protein precipitate was insoluble after processing.

To explore the potential relationship between muscle secretion of myokines and the levels of these peptides released in the circulation, we performed correlation analyses between serum and human primary muscle cell-culture supernatant concentrations. There were no significant correlations between the levels of the candidate myokines in serum and their secretion by the myotubes, suggesting that the peptides quantified in the circulation of the participants are not predominantly originating from muscle tissue. As mentioned previously, serum is not the preferential medium to quantify myokine variations in response to exercise interventions, as differences in their circulatory secretion are better detected in plasma samples [62]. Consequently, there is a possibility that changes in plasma concentrations could have better correlated with myotube secretion of the myokines in response to the exercise intervention. These correlation analyses further highlight the importance of measuring myokine secretion from muscle cells or tissue directly, rather than relying on the assumption that circulating levels of these peptides represent their release from muscle and not other tissues.

The acute muscle contraction in vitro model we employed was validated in a mouse muscle cell line (C2C12 cells) [57] with a very different morphology in comparison to primary human muscle cells. Indeed, C2C12 muscle cells have better contractile activity than human primary muscle cells, as they form more tubular myotubes that better resemble muscle fibers mechanically than the flatter human myotubes [58,59]. Nonetheless, we saw an increase in the phosphorylation of proteins involved in signal transduction of the AMPK pathway in response to the EPS treatment (AMPK, AS160, and ACC). This molecular signaling adaptation was not paralleled by increased myokine secretion by the myotubes as we expected. Others found EPS to significantly increase myokine secretion in human primary muscle cells, but their stimulation protocols were of longer durations [51,52]. Another important consideration in the interpretation of our results is the discrepancy that was observed by others in the response to EPS treatment as a function of BMI and insulin resistance [56]. Indeed, Park et al. showed that myotubes derived from participants with severe obesity (BMI > 40 kg/m^2^) and with whole-body insulin resistance showed reduced activation of the AMPK pathway and downstream GLUT4 translocation to the plasma membrane in response to a 24 h EPS treatment compared to myotubes derived from lean and insulin-sensitive participants. Since participants in our study lived with obesity, and some qualified as having severe obesity, there is a possibility that the participants’ characteristics impacted the ability of the myotubes derived from their skeletal muscle biopsy to respond to EPS similarly to myotubes isolated from participants not living with obesity. As this study did not include a group of participants without obesity, we cannot determine whether this factor played a role in the observed response to EPS treatment in the assessed myotubes. Also, it is possible that the EPS protocol we employed is sufficient to activate some of the pathways induced during muscle contraction but that some other metabolic or mechanical adaptations, such as Ca^2+^ dynamics, are not induced sufficiently to lead to the release of myokines occurring in vivo during muscle contraction. That being said, we did observe mechanical contraction of the myotubes at a varying proportion in the majority of the cell lines during EPS exposure.

Our analysis of the mRNA expression of myokines in the myotubes obtained from biopsies collected from the participants pre- and post-intervention revealed interesting patterns in their regulation. For example, the expression of pro-inflammatory myokines IL-1β and IL-8 were both upregulated in myotubes from the T2D responders group following the aerobic exercise intervention, whereas IL-15 mRNA was increased post-intervention in the myotubes from the no-T2D and T2D non-responders groups. This finding is counterintuitive, as IL-8 and IL-15 are both found to be increasingly secreted by muscle cells derived from patients with T2D [63,64]. One hypothesis for increased myokine secretion in human primary myotubes from patients with T2D is that this dysregulation contributes to the low-grade chronic inflammation state observed in this patient population. Therefore, we would expect that increased exercise in patients with T2D would reduce inflammation and downregulate myokine secretion in their muscles. It is possible that we could have observed a downregulation of the expression of these myokines in the patients with T2D at a time-point of collection of the muscle biopsies later than 10 weeks after the start of the exercise intervention or after a longer exercise intervention. Some metabolic and/or signaling changes require a longer period of adaptation, and a 10-week intervention is a relatively short snapshot of a chronic metabolic condition ongoing for years prior to the lifestyle intervention. It is also possible that alterations in muscle expression of the myokines are not reflective of protein levels measured in the cell-culture supernatant because other post-transcriptional mechanisms affect their accumulation in the medium, such as mRNA translation and autocrine/paracrine myokine uptake by the muscle cells.

The most interesting comparison between the response to the aerobic exercise protocol across all groups was the change in the expression of IL-10 mRNA. In both the myotubes from the no T2D and T2D responders groups, IL-10 mRNA was increased post- compared to pre-intervention, while the opposite could be observed in the myotubes from participants in the T2D non-responders group. IL-10 is an anti-inflammatory myokine that positively modulates glucose metabolism [13], and to our knowledge, this is the first exploration of its regulation at the mRNA level in myotubes from patients with obesity with or without T2D accounting for the ability of patients with T2D to respond to exercise training. Although the statistical analysis of IL-10 secretion in the cell-culture supernatant of the myotubes showed no significant change in response to the exercise intervention, in all study participants irrespective of group, post-intervention concentrations were higher than pre-intervention. Others also found an increase in IL-10 in the plasma of patients with T2D after an aerobic exercise intervention, but their experimental plan did not include the measurement of muscle secretion or expression [65]. Interestingly, Barry et al. found what they described as a hyporesponsiveness to the anti-inflammatory effect of IL-10 in patients with T2D [66]. This altered response to IL-10 in the context of T2D might explain the increase in its secretion from muscle cells in patients with this chronic condition to compensate for reduced sensitivity to the anti-inflammatory effect of this myokine. To determine whether the observed increase in IL-10 secretion from the myotubes was similar amongst the groups, we compared the percentage increase in IL-10 concentrations post- compared to pre-intervention and found no significant differences.

In summary, contrary to our research hypothesis, we found no definitive implications of myokine secretion in the mechanisms of exercise resistance in patients with T2D. However, we did find novel regulatory patterns of mRNA expression in myotubes derived from participants as a function of the exercise intervention and the ability of patients with T2D to respond to exercise training. We also showed that the EPS conditions developed in C2C12 cells to reproduce acute muscle contraction showed great potential as an exercise mimetic in primary human muscle cells, as it activated the AMPK pathway and resulted in visible contraction of the myotubes. Further analyses and observations of the resulting metabolic adaptations in the myotubes must be performed to establish whether this model fully reproduces in vivo muscle contraction. This study highlights that the regulation of local myokine signaling in the muscles of patients with T2D warrants further investigation to better understand the role of these important signaling molecules in the development of muscle metabolic defects in patients with T2D.

## 4. Materials and Methods

### 4.1. Study Design

The participant population in this study was included in a larger clinical trial previously published (NCT01911104). Participants had to be sedentary, which corresponds to not being physically active ≥ 3 days per week for a period of 6 months. Samples from 7 participants with obesity (BMI > 30 kg/m^2^) and 13 participants with obesity and T2D (self-reported or with a fasting glucose *≥* 7 mmol/L) were used for this secondary analysis. The recruited participants underwent an aerobic exercise training intervention for 10 weeks. Participants living with T2D ceased glucose-lowering treatment 15–17 days prior to the start and for the duration of the exercise intervention. Fasting blood samples were collected 72 h before the start of the intervention (day −3) and seven days after the last exercise session (day 77). In vivo muscle mitochondrial function was assessed via phosphorus (^31^P) magnetic resonance spectroscopy with a 3-T Achieva magnet (Philips Healthcare, Andover, MA, USA) in the patients with T2D at day −3 and 77 to differentiate the responders (increase in PCr recovery rate) from the non-responders (decrease in PCr recovery rate) to the exercise intervention as described previously [42]. Insulin sensitivity (hyperinsulinemic–euglycemic clamp), maximal aerobic capacity (VO_2peak_; incremental treadmill test on a Trackmaster TMX 425c (Full Vision, Inc., Newton, KS, USA)) and body composition (dual X-ray absorbsiometry (Lunar iDXA Whole-body Scanner, GE Healthcare Lunar, Madison, WI, USA) were also assessed at day −3 and 77. A skeletal muscle biopsy was performed with a Bergström needle in the vastus lateralis at day −3 and 77 of the exercise intervention for a sub-group of participants (3 no T2D, 6 T2D responders, and 3 T2D non-responders).

### 4.2. Exercise Intervention Protocol

The intervention consisted of 4 sessions per week of supervised aerobic exercise on a treadmill following a ramped protocol over 10 weeks [41]. The intensity was monitored as a function of the achieved target heart rate (HR). During weeks 1–4, participants exercised for at least 20 min at an intensity corresponding to 50–70% of their VO_2peak_, increasing to 45 min at the same intensity for weeks 5–8, and for weeks 9–10, participants exercised 45 min at 75% of their VO_2peak_. The adherence to the exercise intervention protocol was measured as the time spent at their target HR during the exercise sessions as a function of the total duration of the sessions.

### 4.3. Cell Culture, Electrical Pulse Stimulation, and Preparation of Cell Lysates

Primary human muscle cells were isolated from a fraction of the muscle biopsy and cultured in vitro, then purified using the mouse monoclonal 5.1H11 anti-CD56 antibody as previously described [67]. Purified myocytes were grown in Ham-F10 supplemented with 10% fetal bovine serum (FBS), 1 µM dexamethasone, 10 ng/mL epidermal growth factor, 25 pmol/L insulin (from bovine pancreas, MilliporeSigma, Oakville, ON, Canada), 1X antibiotic/antimycotic, and 5 µg/mL gentamycin (all cell-culture reagents were from Wisent, Saint-Jean-Baptiste, QC, Canada unless otherwise stated) in matrigel-coated (Corning, New-York, NY, USA) T-75 flasks. At passages 4–5, cells were plated in matrigel-coated 6-well plates (Corning, New-York, NY, USA) suitable to fit with the carbon electrode of the Ion-Optix C-Pace electrical pulse stimulation (EPS) apparatus and grown to confluency before the start of differentiation into myotubes in DMEM low glucose (5 mmol/L; Wisent, Saint-Jean-Baptiste, QC, Canada) supplemented with 2% FBS, 1X antibiotic/antimycotic and 5 µg/mL gentamycin for seven days. During the last 24 h of differentiation, cells were changed to a phenol-red-free differentiation medium (Wisent, Saint-Jean-Baptiste, QC, Canada). Myotubes were then treated with three different stimulation conditions for 1 h in fresh serum-free phenol-red-free DMEM low glucose supplemented with 1X antibiotic/antimycotic and 5 µg/mL gentamycin: non-stimulated (NS), electrode exposure (E) (6-well carbon electrode on the cells with no current), and stimulated (S) with 20 V 24 ms/s impulses at 1 Hz as in [57]. The cell-culture supernatants were collected, centrifuged at 2000× *g* for 10 min to remove cell debris, and kept at −80 °C for myokine quantification. The cells were harvested in Trizol^®^ (Invitrogen, Waltham, MA, USA) and kept at −80 °C for mRNA isolation. The same EPS protocol was repeated in all cell lines using phenol-red-free differentiation medium (DMEM low glucose with 2% FBS) for immunoblot analyses, with cells lysed in a buffer containing 50 mM Tris (pH 8.0), 0.1% Nonidet P-40 (NP-40), 0.5 mM dithiothreitol (DTT), 10 mM sodium fluoride (NaF), 1 mM sodium orthovanadate (Na_3_VO_4_) (all from MilliporeSigma, Oakville, ON, Canada), and Pierce proteinase inhibitors tablet (A32963, Thermofisher, West Etobicoke, ON, Canada). The lysates were sonicated and centrifuged at 12,000 rpm for 10 min at 4 °C, and the supernatants were stored at −80 °C.

### 4.4. Immunoblotting

AMP-activated protein kinase (AMPK) pathway activation was assessed in all cell lines pre- and post-intervention in response to the EPS treatments by polyacrylamide gel electrophoresis. Clarified lysates were prepared from myotubes submitted to EPS treatments in the presence of 2% FBS, as described above, to prevent the confounding effects of serum starvation on AMPK-pathway activation. Cell lysates were separated in Laemmli buffer on 8% SDS-polyacrylamide gels by electrophoresis and transferred to polyvinylidene difluoride (PVDF) membranes for immunoblotting. The primary antibodies used were P-AMPKα (Thr172) (#2535), AMPKα (#5831), P-ACC (Ser79) (#11818), ACC (#3676), P-AS160 (#4288), AS160 (#2447), diluted 1:500, and *α*-tubulin (#2144) (Cell Signaling Technologies, Waltham, MA, USA) diluted 1:1000 all in TBS-Tween 0.1%–BSA 5%. The secondary antibodies used were anti-rabbit or anti-mouse coupled to horseradish peroxidase diluted 1:5000 in TBS-Tween 0.1%–milk 5% (Santa Cruz Biotechnology, Dallas, TX, USA). Proteins were visualized using chemiluminescent substrates on the chemiSOLO (Azure Biosystems, Dublin, CA, USA), and protein bands were quantified by integrated density analysis in ImageJ2 software version 2.14.0 (National Institutes of Health, Bethesda, MD, USA).

### 4.5. Quantification of Myokine Secretion

Myokines were measured in serum samples collected from the participants at baseline and follow-up of the exercise intervention, as well as in the cell-culture supernatants of myotubes derived from biopsy samples obtained at baseline and follow-up timepoints. Myokine quantifications were performed using a single-plex assay (R-plex; SPARC) and a multiplex assay (U-plex; BDNF, FGF21, IL-6, IL-8, IL-10, IL-13, IL-15, and IL-18) from Meso Scale Discovery (Rockville, MD, USA) following the manufacturer’s instructions and as previously published [68,69]. All antibodies with the exception of SPARC were validated for target specificity. More information about specificity tests can be found in the datasheet of the U-plex kits (https://www.mesoscale.com/ (accessed on 21 March 2024)).

### 4.6. Quantification of mRNA Expression

Cell lysates harvested in Trizol^®^ (Invitrogen, Waltham, MA, USA) were processed following the Trizol mRNA isolation protocol. Briefly, chloroform was added to the lysates in a 1:5 (*v*:*v*) ratio and the samples were vortexed, incubated on ice, and centrifuged at 12,000 rpm for 15 min at 4 °C. The upper (aqueous) phase was then collected and mixed with ice-cold isopropanol before centrifuging at 12,000 rpm for 10 min at 4 °C to precipitate the RNA. The pellet was dried and washed in 70% ethanol before storage at −20 °C until the day of the assay. On the day of the NanoString assay, the samples were thawed on ice, centrifuged at 12,000 rpm for 10 min at 4 °C, and dried completely before resuspending the pellet in RNase-free water. RNA samples were measured with the Nanodrop, and the concentrations were adjusted for downstream analyses. Expression patterns of myokines and other genes of interest were assessed with the nCounter^®^ Human Inflammation V2 Panel from NanoString Technologies (Seattle, WA, USA). The samples were processed according to the manufacturer’s instructions, and data were analyzed using the n-Solver Analysis Software V4.0 (NanoString Technologies, Seattle, WA, USA).

### 4.7. Statistical Analyses

All patient characteristics were first assessed for normality or lognormality using the D’Agostino and Pearson test. Datasets that followed a lognormal distribution were transformed before further analyses. The age and adherence data were analyzed using an ordinary ANOVA with Šidák’s multiple comparison as a post hoc test. The rest of the patient characteristics pre- (day −3) and post-intervention (day 77) (BMI, HbA1c (%), fasted blood glucose (mg/dL), fat mass (kg), lean mass (kg), M-value (mg/kg/min), and VO2_max_ (mL/kg/min)). The mRNA quantification assays were analyzed by two-way ANOVA with repeated measures or mixed-model analysis when some values were missing with Šidák’s multiple comparison as a post hoc test. The quantification data of the immunoblots were analyzed by regular two-way ANOVA with Šidák’s multiple comparison as a post hoc test. Data relating to myokine quantification were first analyzed to identify potential outliers by the ROUT method using a Q = 1%. The ‘cleaned’ data were then analyzed by two-way ANOVA with repeated measures or mixed-model analysis when some values were missing with Šidák’s multiple comparison as a post hoc test. Correlation analyses of myokines measured in the cell-culture supernatant and the serum samples were performed by pooling pre- and post-intervention values from participants across all groups. Data were first assessed for normality using the D’Agostino and Pearson test. Data sets that did not follow a normal distribution were analyzed using the nonparametric Spearman correlation test, whereas those following a normal distribution were assessed using Pearson correlation coefficients. A *p*-value *≤* 0.05 was considered as significant. Statistical analyses were performed using Prism 10.2.1 Software (Graphpad, Boston, MA, USA).

## Figures and Tables

**Figure 1 ijms-25-04889-f001:**
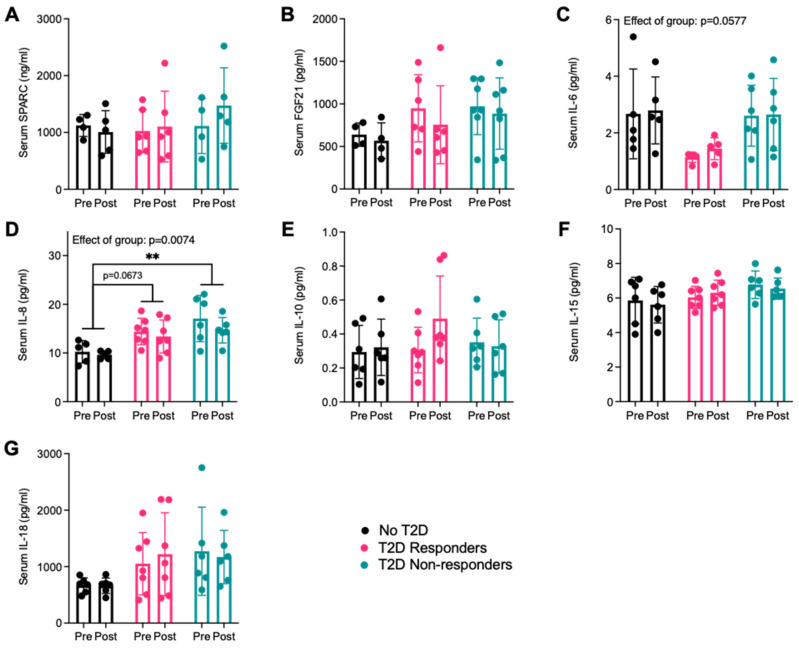
Variations in resting cytokine serum concentrations following the 10-week aerobic exercise intervention. (**A**) SPARC, (**B**) FGF21, (**C**) IL-6, (**D**), IL-8, (**E**) IL-10, (**F**) IL-15, and (**G**) IL-18 levels are presented before (pre) and after (post) the exercise intervention. IL-13 was undetected in almost all serum samples. No T2D = participants with obesity without T2D, T2D responders = patients with T2D who respond to training, and T2D non-resp. = patients with T2D who do not respond to training. N = 4–7. Data are represented as individual values and mean ± SD. ** *p* < 0.01.

**Figure 2 ijms-25-04889-f002:**
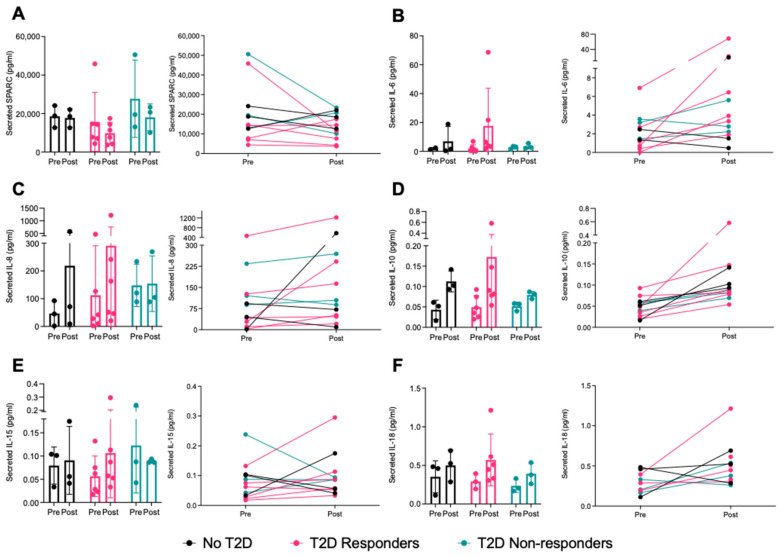
Variations in myokine secretion in cell-culture supernatant of human primary myotubes in response to the 10-week aerobic exercise intervention. Myokines were measured by R-plex and U-plex assays (MSD, Maryland, USA). (**A**) SPARC, (**B**) IL-6, (**C**) IL-8, (**D**) IL-10, (**E**) IL-15, and (**F**) IL-18 levels are presented before (pre) and after (post) the exercise intervention. BDNF, FGF21, and IL-13 were mostly undetected. No T2D = participants with obesity without T2D, T2D responders = patients with T2D who respond to training, and T2D non-resp. = patients with T2D who do not respond to training. N = 2–6. Data are presented as individual data points with mean ± SD and line graphs when applicable.

**Figure 3 ijms-25-04889-f003:**
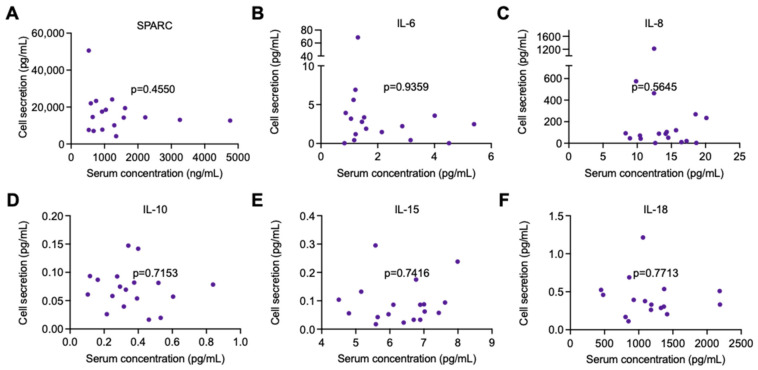
Correlation analysis between myokine secretion in cell-culture supernatant of human primary myotubes and serum concentrations of cytokines from the participants. Myokines were measured by R-plex and U-plex assays (MSD, Maryland, USA). (**A**) SPARC, (**B**) IL-6, (**C**) IL-8, (**D**) IL-10, (**E**) IL-15, and (**F**) IL-18 levels. N = 16–18. Data are presented as individual data points including pre- and post-intervention values per participant.

**Figure 4 ijms-25-04889-f004:**
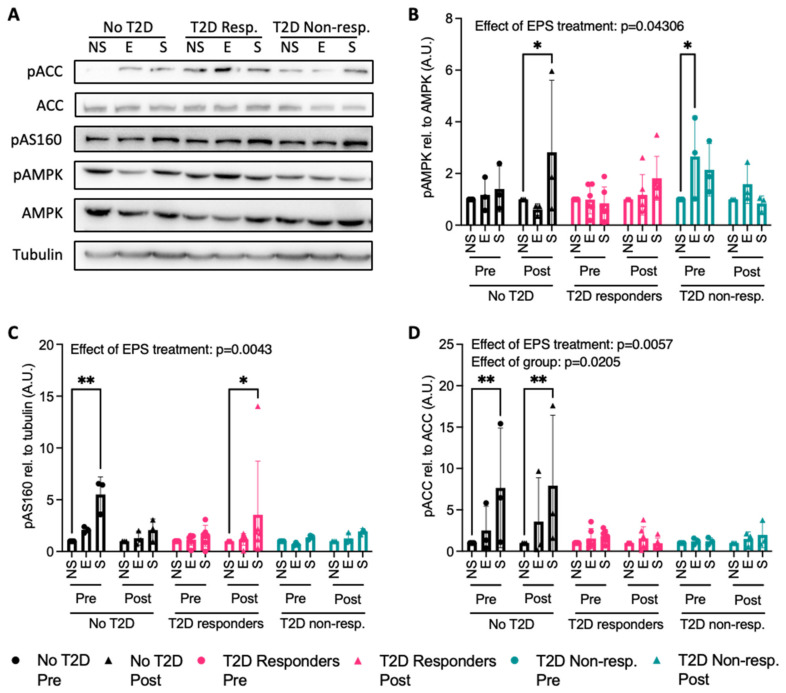
Effect of EPS on the phosphorylation of proteins involved in the AMPK pathway in human primary myotubes quantified by immunoblotting. NS = non-stimulated, E = electrode exposure, S = EPS treatment, Pre = pre-intervention, Post = post-intervention. (**A**) Representative blot, (**B**) quantification of p-AMPK relative to total AMPK, (**C**) quantification of p-AS160 relative to tubulin, and (**D**) quantification of p-ACC relative to total ACC. Blots were analyzed with ImageJ (NIH). No T2D = participants with obesity without T2D, T2D responders = patients with T2D who respond to training, and T2D non-resp. = patients with T2D who do not respond to training. N = 3–6. Data are shown as mean ± SD with individual data points. * *p* < 0.05, ** *p* < 0.01.

**Figure 5 ijms-25-04889-f005:**
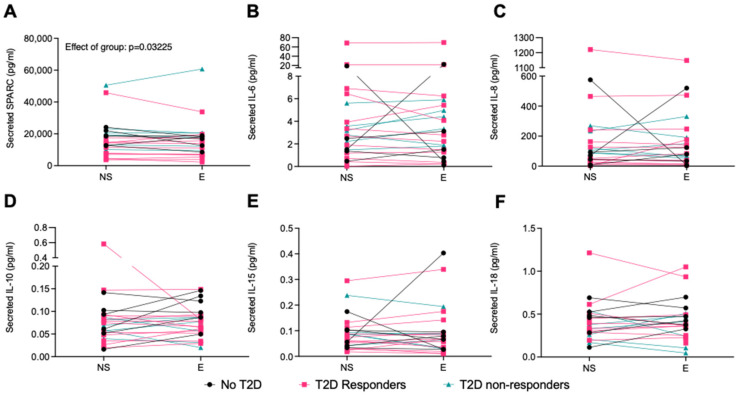
Effect of electrode-exposure control condition of human primary myotubes collected pre- and post-intervention on myokine secretion. Myokines were measured in the cell-culture supernatant by R-plex and U-plex assays (MSD, Maryland, USA). (**A**) SPARC, (**B**) IL-6, (**C**) IL-8, (**D**) IL-10, (**E**) IL-15, and (**F**) IL-18. BDNF, FGF21, and IL-13 were mostly undetected. No T2D = participants with obesity without T2D, T2D responders = patients with T2D who respond to training, and T2D non-resp. = patients with T2D who do not respond to training. Non-stimulated = NS, exposed to the electrode apparatus for 1 h = E. N = 6–12. Data are shown individually with lines connecting the NS and E conditions for one cell line.

**Figure 6 ijms-25-04889-f006:**
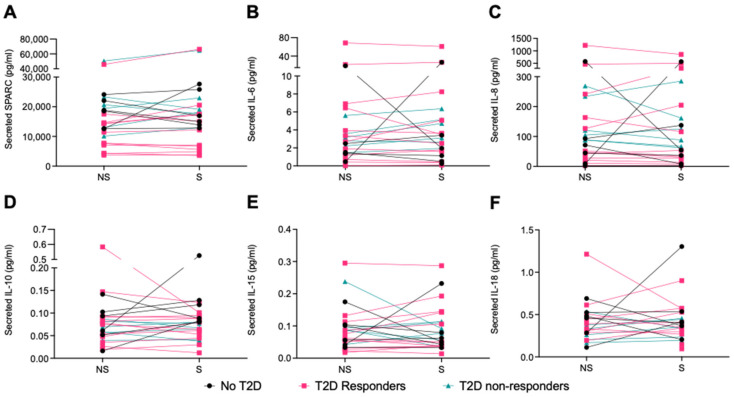
Effect of EPS treatment of human primary myotubes collected pre- and post-intervention on myokine secretion. Myokines were measured in the cell-culture supernatant by R-plex and U-plex assays (MSD, Maryland, USA). (**A**) SPARC, (**B**) IL-6, (**C**) IL-8, (**D**) IL-10, (**E**) IL-15, and (**F**) IL-18. BDNF, FGF21, and IL-13 were mostly undetected. No T2D = participants with obesity without T2D, T2D responders = patients with T2D who respond to training, and T2D non-resp. = patients with T2D who do not respond to training. N = 6–12. Data are shown individually with lines connecting the non-stimulated (NS) and EPS-stimulated (S) conditions for one cell line.

**Figure 7 ijms-25-04889-f007:**
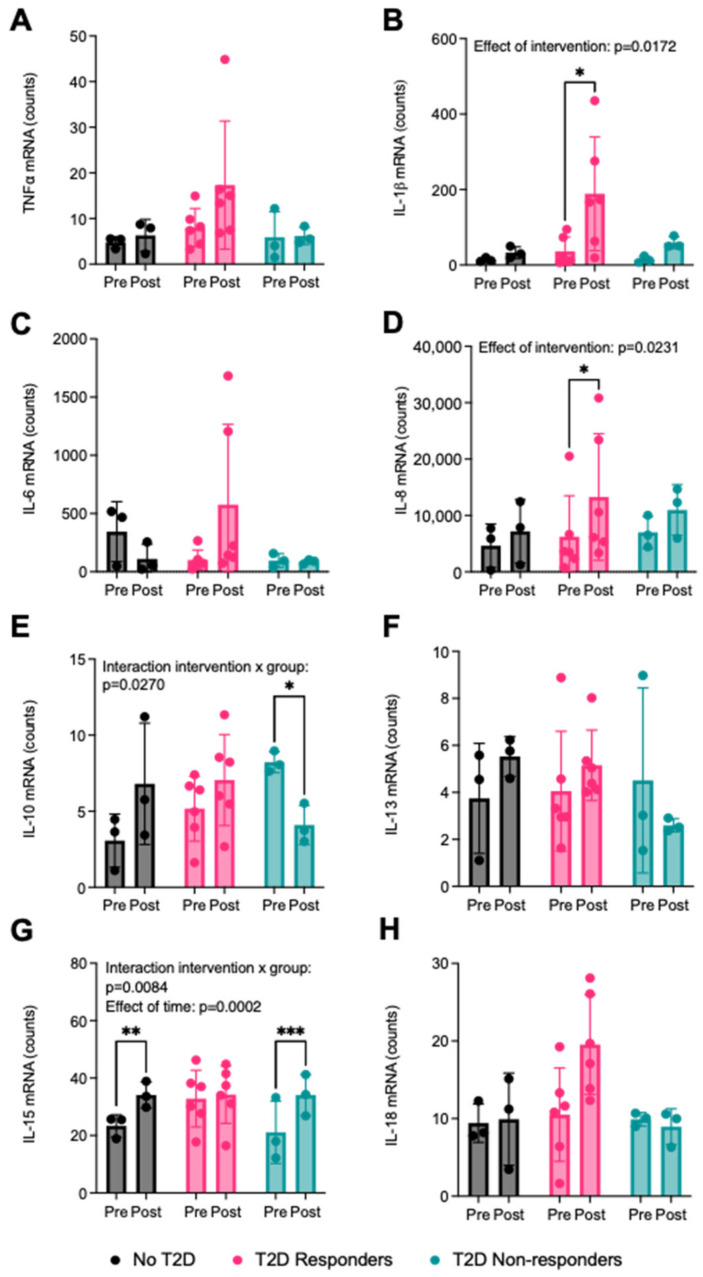
Effect of the 10-week aerobic exercise intervention on the mRNA expression of the mRNA expression of myokines in human primary myotubes as assessed by NanoString multiplex assay. Data analysis was performed with nSolver Analysis Software v4.0 and individual quantification (counts) of mRNA molecules were extracted for targets thought to be candidate myokines. Relative counts of (**A**) TNF-α, (**B**) IL-1β, (**C**) IL-6, (**D**) IL-8, (**E**) IL-10, (**F**) IL-13, (**G**) IL-15, and (**H**) IL-18 quantified by multiplex assay before (pre) and after (post) exercise intervention. No T2D = participants with obesity without T2D, T2D responders = patients with T2D who respond to training, and T2D non-resp. = patients with T2D who do not respond to training. N = 3–6. Data are shown as mean ± SD with individual data points. * *p* < 0.05, ** *p* < 0.01, *** *p* < 0.001.

**Table 1 ijms-25-04889-t001:** Participant characteristics pre- and post-10-week aerobic exercise intervention.

Group	n (m/f)	Age (Years)	BMI (kg/m^2^)		HbA1c (%)		Fat Mass (kg)		Lean Mass (kg)		M-Value (mg/kg/min)		VO_2max_ (mL/kg/min)		PCr Recovery Rate (1/s)	
			Pre	Post	Pre	Post	Pre	Post	Pre	Post	Pre	Post	Pre	Post	Pre	Post
No T2D	7 (2/5)	45.3 ± 6.3	37.2 ± 6.7	37.4 ± 5.9	5.5 ± 0.5	5.7 ± 0.4	44.1 ± 12.0	44.5 ± 11.3	51.2 ± 12.8	51.9 ± 12.2	168.0 ± 95.3	231.4 ± 116.5	22.1 ± 6.1	23.6 ± 5.4	0.025 ± 0.014	0.022 ± 0.008
T2D responders	9 (6/3)	53.4 ± 7.1	34.8 ± 3.6	34.6 ± 3.9	7.6 ± 0.8 ^†††^	7.9 ± 0.9 ^†††^	40.2 ± 8.9	39.7 ± 9.4	58.7 ± 10.0	59.1 ± 9.7	59.8 ± 51.5 ^†^	78.6 ± 43.1 ^††^	23.2 ± 4.1	25.1 ± 5.1	0.015 ± 0.003 ^‡‡^	0.020 ± 0.004
T2D non-responders	6 (3/3)	47.2 ± 9.5	34.4 ± 4.5	34.8 ± 4.7	7.4 ± 0.9 ^†^	7.9 ± 1.5 **^†††^	39.5 ± 9.3	40.6 ± 9.8	54.4 ± 13.9	56.0 ± 14.2 **	114.2 ± 54.5	99.7 ± 71.9 ^††^	24.1 ± 7.5	22.7 ± 4.2	0.026 ± 0.005	0.020 ± 0.004 **

Data are shown as mean ± SD. ** *p* < 0.01 in comparison to pre-intervention. ^†^
*p* < 0.05, ^††^
*p* < 0.01, and ^†††^
*p* < 0.001 in comparison to no T2D group at the same time point of intervention. ^‡‡^
*p* < 0.01 compared to the non-responders group at the same time point of intervention. No T2D = participants with obesity without T2D, T2D responders = patients with T2D who respond to training, and T2D non-resp. = patients with T2D who do not respond to training. m = males. f = females. BMI = body mass index. HbA1c = percentage glycated hemoglobin. M-value = whole-body insulin sensitivity. VO_2max_ = maximal oxygen consumption rate. PCr = phosphocreatine.

## Data Availability

The raw data supporting the conclusions of this article will be made available by the authors on request.

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
