# Peer review of "Myokine Secretion following an Aerobic Exercise Intervention in Individuals with Type 2 Diabetes with or without Exercise Resistance"

_ijms, 2024, doi:10.3390/ijms25094889_

Round 1

Reviewer 1 Report

Comments and Suggestions for Authors

Please see comments in attached file.

Reviewer 2 Report

Comments and Suggestions for Authors

I have read and analyzed the manuscript from Garneau and coauthors. In my opinion the manuscript is devoted to important and prospective problem. However, the manuscript did not contain Figures 5 and 7 and clear understanding of author's conclusions and ideas is effortful. After addition of these figures I can consider the manuscript completely. Nevertheless, I have a couple of questions to the first version.

1.My suggestion is transfer Table 1 into Supplement and make Figure 1 with crucial box and whisker plots.

2.Table 1, Can authors present M-index values?

3.Figure 4. Why authors did not use tAS160 antibody for the quantification? It would be more relevant.

4.Authors have primary myotubes from patients. Why electrostimulation has been used for the C2C12 cell line, but not for primary cells? The results on primary cells would be more relevant.

Reviewer 3 Report

Comments and Suggestions for Authors

The authors examined the impact of aerobic training on myokine expression in obese subjects without T2D as well as in T2D exercise responders and no responders. Different myokines were measured in serum as well as in cell supernatants of biopsy-derived primary muscle cells (in these mRNA levels were also quantified) before and after exposure to exercise. The authors also examined the involvement of the AMPK- signaling pathway.

The study showed no effect of either T2D or the response to an exercise intervention at the protein, however, on mRNA (for some myokines) level of the measured myokines.

Major:

My major concern is very low sample size. I’m wondering whether the authors performed power calculation and whether the lack of significant effects (group differences and correlations) might be due to low sample size.

Correlation analyses across all groups pre- and post-intervention is not optimal, however possible only in this way due to low N.

The authors acknowledged several important limitations, except low sample size.

Minors:

Lane 170: ..in the T2D non-responders; the fig shows a trend in responders

ACC and pAS160 are not explained.

Fig. 7J: as it stands is not informative as the font is too small.

Lane 421: post-translational should be replaced with post-transcriptional

Round 2

Reviewer 2 Report

Comments and Suggestions for Authors

Many thanks to the authors for the comprehensive response. The manuscript can be accepted for publication.

Reviewer 3 Report

Comments and Suggestions for Authors

The authors addressed all the points raised by the reviewer and improved the manuscript.